# Information Fusion-Based Deep Neural Attentive Matrix Factorization Recommendation

**Zhen Tian** [1,2] (ID)**, Lamei Pan** [2]**, Pu Yin** [1,2] **and Rui Wang** [1,2,*]

1   Shunde Graduate School, University of Science and Technology Beijing, Foshan 528300, China; tian_zhen1024@163.com (Z.T.); g20208848@xs.ustb.edu.cn (P.Y.)
2   School of Computer & Communication Engineering, University of Science and Technology Beijing, Beijing 100083, China; lamei_pan@163.com
*   Correspondence: wangrui@ustb.edu.cn

**Abstract:** The emergence of the recommendation system has effectively alleviated the information overload problem. However, traditional recommendation systems either ignore the rich attribute information of users and items, such as the user's social-demographic features, the item's content features, etc., facing the sparsity problem, or adopt the fully connected network to concatenate the attribute information, ignoring the interaction between the attribute information. In this paper, we propose the information fusion-based deep neural attentive matrix factorization (IFDNAMF) recommendation model, which introduces the attribute information and adopts the element-wise product between the different information domains to learn the cross-features when conducting information fusion. In addition, the attention mechanism is utilized to distinguish the importance of different cross-features on prediction results. In addition, the IFDNAMF adopts the deep neural network to learn the high-order interaction between users and items. Meanwhile, we conduct extensive experiments on two datasets: MovieLens and Book-crossing, and demonstrate the feasibility and effectiveness of the model.

**Keywords:** attention mechanism; cross-features; deep neural network; information fusion; matrix factorization; recommendation system

## 1. Introduction

In recent years, the numbers and types of data sources, such as internet users and applications, have increased rapidly, leading to an exponential explosion of internet information. Huge volumes of data are created, collected, and processed [1]. This has resulted in a huge contradiction between the massive data supply and the personalized needs of users. The massive dataset makes it difficult for users to quickly obtain information that meets their individualized needs, which is known as information overload [2]. How to balance the massive data with the user's personalized needs, and quickly and accurately provide users with personalized information from the massive data has become an urgent problem to be addressed. The recommendation system is considered to be an effective solution to alleviate this problem. By analyzing users' preferences, the recommendation system can actively recommend information to users that meet their interests [3]. However, traditional recommendation systems usually adopt a single-form data, for instance, the matrix factorization-based recommendation system is often based on the factorization of the rating matrix composed of explicit feedback data or the interaction matrix composed of implicit feedback data to complete the recommendation task. Adopting single-form input data has insufficient information and is easily affected by noisy data [4,5], so it is hard to accurately and comprehensively model users and items, and it is difficult to meet the user's needs for the novelty of the item, which limits the performance of the recommendation systems. In fact, in addition to the rating information, there is other auxiliary information, such as the user's social-demographic features (age, gender, occupation, etc.), the item's content

features (category, introduction, etc.), etc. By adopting information fusion technology [6], the recommendation system could integrate the auxiliary information into the modeling process of the recommendation system, so the recommendation system could model user preference and items representation more comprehensively, which could enhance the expressive ability of the model and improve the performance of recommendations, as well as increase the novelty of the recommendation results.

Based on the characteristic that deep neural network can fuse arbitrary continuous features and category features, some recommendation systems try to introduce auxiliary information into matrix factorization-based recommendation system. By utilizing neural networks, the recommendation system can extract feature representations of auxiliary information and combine them with the matrix factorization. It improves the model's expression ability. However, when fusing the feature representations extracted from the auxiliary information, most of models utilize the fully connected networks to combine and transform these features, which combines different categories of feature vectors by simple splicing. We can generally understand the method that the fully connected network processes feature vectors as addition operations, and addition is equivalent to the "or" relationship in logic. Therefore, such a feature fusion method can be simply and intuitively understood as an "or" operation. However, the "or" operation ignores the interaction between different feature domains. In practical applications, the interaction of features, that is, the "and" relationship between features, often contains higher-value information. Therefore, other feature combination methods are requested to introduce the "and" relationship between features.

In this paper, we propose the information fusion-based deep neural attentive matrix factorization (IFDNAMF) recommendation model. We introduce the auxiliary information to the model based on the feature that deep neural network can fuse arbitrary continuous features and category features. When conducting information fusion, to complete the interactive fusion between features of different information domains and to get different cross-features in a targeted way, we adopt the element-wise product. Meanwhile, to distinguish the importance of diverse cross-features on recommendation results, we also introduce the attention mechanism to learn the weights of different cross-features.

Furthermore, the recommendation system based on matrix factorization suffers from the limitation of the simple linear inner product. In addition, during the inner product of traditional matrix factorization, the results of all dimensions are accumulated with the same weight, which could be seen as the connection weights all being 1, to get the final scalar result. Both of these make it non-effective to model the complex non-linear relationships between the user and the item. To solve these ends, He et al., proposed the generalized matrix factorization (GMF) [7], which utilizes the neural network to endow MF with non-linear learning capability through the activation function and combines the values of different dimensions with different weights by introducing weight matrix. GMF is empowered with the ability to model the second-order non-linear interaction between users and items. The expression ability of the model gets significant improvement. Despite all these advantages, it is not effective for GMF to capture the high-order interaction between users and items that contains richer information because of the shallow network structure, which may limit the performance of the recommendation system. Therefore, hidden layers are introduced on the second-order interaction obtained by the non-linear element-wise product in IFDNAMF, by the deep neural network, we could model the complex non-linear high-order interaction between the user and the item.

The main contributions of our proposed approach can be summarized as follows.

1. We first propose a new recommendation model, the information fusion-based deep neural attentive matrix factorization (IFDNAMF) recommendation model, in which we introduce the auxiliary information to assist the model in describing the user features and item features more comprehensively and specifically;

2. Then, we propose a new method of information fusion, in which the inner product between the different information domains is adopted to learn the cross-features,

so that the IFDNAMF could obtain the "and" relationship containing the higher-value information between the auxiliary information. Meanwhile, to distinguish the importance of diverse cross-features on recommendation results, we also introduce the attention mechanism to learn the weights of different cross-features;

3. Finally, we conduct extensive experiments on two datasets: MovieLens and Book-crossing. The experimental results demonstrate the outstanding performance of IFDNAMF.

## 2. Related Work

The key of the recommendation system is the recommendation algorithm. There are mainly three recommendation algorithms: content based [8], collaborative filtering [9], and hybrid based [10], among which collaborative filtering is the most commonly used. The matrix factorization (MF) algorithm proposed by Simon Funk (Available online: http://sifter.org/simon/journal/20061211.html (accessed on 26 September 2021)) is an effective collaborative recommendation algorithm, which gets widely used in many real-world scenarios [11]. The traditional recommendation system based on matrix factorization decomposes the user rating matrix into the user-factor matrix and the item-factor matrix through a set of potential feature factors and utilizes potential factor vectors to describe users and items, that is, users and items are mapped into a shared latent vector space, and the interaction between users and items is expressed by the inner product between the mapped vectors, and then a Top_N recommendation list is generated for the user according to the inner product result [12].

However, the traditional matrix factorization-based recommendation system often takes the rating matrix or interaction matrix as the input. The single-form input data contain a small amount of information, thus it is difficult to model user preference and item representation comprehensively and accurately. Furthermore, MF faces the problem of high sparsity of rating matrices and interaction matrices since each user only interacts with a tiny fraction of a great quantity of items, which limits the performance of recommendation systems [13]. In fact, in practical application scenarios, users and items have a wealth of auxiliary information, such as social trust relationships [14,15], tags [16], context [17], and so on. Introducing the auxiliary information to the model can assist the recommendation system in modeling user preferences and item representations more comprehensively, and further improve the model's performance of recommendation systems. Therefore, to make full use of the rich auxiliary information to assist in modeling user preferences and item representations, Yuka Wakita et al. [18] employed the auxiliary information in the recommendation model to solve users' brand recommendation task. A large number of user and item features are modeled in the model, including the user's age, acceptable price range, item materials, design methods, sewing methods, etc. These learned feature representations are mapped to the softmax output layer through multi-layer neurons with ReLU as activation function to obtain predicted ratings. Similarly, Yi et al. [19] proposed a deep learning based collaborative filtering framework, namely, deep matrix factorization (DMF), which integrates various side information to generate latent factors of users and items by concatenation operation. P Covington et al. [20] proposed a recommendation algorithm based on neural networks, in which multiple fully connected layers are utilized to integrate rich continuous features and category features, including historical user activity records and social-demographic features (age, gender, occupation, etc.), to improve performance of recommendation systems of the model. Furthermore, Greg Zanotti et al. [21] employed the neural network language model to fuse information from multiple sources, including user data, item data, and tag data, to extract rich feature representations of users and items.

The above researches introduce a great quantity of auxiliary information with various forms to the model through the deep neural network, including continuous features and category features, to model user preferences and item representations more comprehensively and accurately. However, when fusing the features of various information, most of these researches utilize the traditional fully connected network to simply concatenate the features to obtain the implicit feature representations of users and items. In brief,

we can understand this combination mode as "or" in logic. This approach has certain limitations and it is not sufficient to model the interaction between features. In other words, the fully connected network cannot obtain cross-features, and cannot express the "and" relationship between different features, but in practical applications, the "and" relationship between features often contains high-value information, efficiently utilizing this information could assist the model in further improving the performance of recommendation systems. Similarly, Shen et al. [22] propose a recommendation framework, namely DVMF, which introduces the implicit feedback information and side information to model low-dimensional feature representation of users and items and greatly improves the recommended performance. However, DVMF still combines the features of side information by splicing, which limits the model performance. Based on the MF and the LSTM, Sun et al. [23] proposed a novel probability framework, named as joint matrix factorization (JMF), which can effectively extract side information to form latent vectors. It also concates the features of the side information by splicing and ignores the interaction of the feature of side information. In addition, Ji et al. [24] propose a hybrid recommendation model based on user ratings, reviews and social data. Social relationship and reviews information are used as auxiliary information to improve recommendation performance. Zafran Khan et al. [25] propose a context-based recommendation model to improve item feature extraction, which extracts context features through convolutional neural network (CNN), it not only resolve the sparsity problem, but also addresses the information loss due to the negative values in latent factors. Although both of them introduce auxiliary information to improve recommendation performance, they ignore the interaction between different features. Of course, some works think about the interaction between the features of side information. For instance, in fusing auxiliary information, Zhao et al. [26] take into account capturing the correlation information between different types of side information, they put forward the HAF framework, which using heterogeneous network to unified model various side information and capture the interaction relationship between different types of side information. To some extent, the performance of recommendation system gets improved. However, when combining features, they still adopt simple splicing, so the side information still cannot be fully utilized to improve performance. In addition, the method generating metagraphs by hand-crafting requires a lot of labor costs and recommendation performance is easily affected by experience. Therefore, the IFDNAMF proposed in this paper introduces a large amount of auxiliary information into the matrix factorization-based recommendation model. When conducting the auxiliary information fusion, a targeted neural network structure is adopted to complete the cross fusion of information of different feature domains in a targeted manner, by which the recommendation system could model user preferences and item representations more comprehensively and accurately.

In addition, the performance of traditional recommendation systems based on matrix factorization is limited by simple linear inner product, although some recommendation tasks [27–30] have been done to improve the matrix factorization model of recommendation system in different directions, which improves the model's performance of recommendation systems, the performance of recommendation model is still limited by employing the inner product to model the interaction between users and items. The linear product may not be sufficient to capture the non-linear structures of interaction between users and items. That is, the simple and fixed inner product has its limitations, which will cause a limitation to the model. To solve this problem, He et al. [7] applied the neural network in the MF model to learn the non-linear interaction between the user and the item according to the non-linear learning ability of the neural network, which greatly improves the performance of recommendation systems of the model. However, GMF only contains the embedding layer and the output layer, which only models the second-order interaction between the user and the item and cannot well capture the high-order interaction which hides richer information. In other words, GMF is a shallow model, which can effectively model the low-level features of interaction between users and items, but it is not sufficient to capture

the high-order interaction that contains a lot of richer and more abstractive information, which may limit the performance of recommendation systems of the model.

In recent years, more and more works have attempted to combine matrix factorization with deep neural network for recommendation tasks because of the ability of the deep neural network to learn high-level and more abstractive features. The work of Aäron van den Oord [31] combined matrix factorization with deep convolutional neural networks. Zhang et al., utilized heterogeneous network embedding and deep learning embedding methods to automatically extract semantic representations from structural knowledge, text knowledge, and visual knowledge in the knowledge base, and then combined them with the matrix factorization model in collaborative filtering to make recommendations [32]. Wang et al. [10] proposed the CDL model, which has good expressive power by performing deep representation learning on content information and collaborative filtering on the rating matrix. Yan et al. [33] proposed an asymmetric neural matrix factorization recommendation system. By adopting the deep neural network, the model could learn the high-order embedding vector representations of users and items from the interaction matrix and then complete the score prediction based on matrix factorization. These methods utilize deep neural networks to learn high-order features of users and items, which endows the model the ability to learn the interaction of high-order abstract features of users and items. However, for the part of collaborative filtering, they still apply MF which combines the feature vectors of users and items by simple linear inner product, it limits the performance of recommendation systems of the model to a certain extent. Therefore, in IFDNAMF recommendation model, by adding hidden layers on the non-linear second-order interaction, IFDNAMF could utilize deep neural networks to model non-linear high-order interaction between users and items, as well as solve the problem that is generated by the simple linear inner product of MF.

## 3. Preliminary Work

In this section, we first present the IFDNAMF recommendation model's framework and describe how it works. Then, we elaborate how our proposed IFDNAMF address the limitations of the existing methods, namely completing feature crosses between different auxiliary information feature domains to model the user preference and item representation more comprehensively, adopting the attention mechanism to learn the importance of different cross-features when conducting information fusion, and modeling the second-order and non-linear high-order interaction by deep neural network.

### 3.1. An Overview of IFDNAMF Framework

To make full use of the auxiliary information to model user preference and item representation more comprehensively and accurately, and, at the same time, learn the complex non-linear high-order interaction between users and items, and then further improve the recommendation performance, this paper proposes information fusion-based deep neural attentive matrix factorization recommendation model, in which, when fusing information, element-wise product operation is employed to complete the interaction between features of different information domains in a targeted manner. In addition, the IFDNAMF adopts the attention mechanism to distinguish the importance of different cross-features on the final latent feature representation during the fusion process. Furthermore, the deep neural network is utilized to model the complex non-linear high-order interactions between users and items. Figure 1 shows the model framework. The framework is mainly composed of 6 parts: input layer, interaction layer, pooling layer, GMF layer, hidden layer, and output layer. The input layer obtains the continuous features and category features of the user and the item, including the user ID, item ID, and other attribute features, the embedding vectors of the user and the item are obtained by embedding technology, and then the cross-features are captured by pairwise element-wise product operation between these embedding vectors. After that, the cross-features are entered into the pooling layer based on the attention mechanism to fusion by different weights learned from the attention

mechanism to obtain the latent features of the user and the item, and then the IFDNAMF models the second-order interaction between the user and the item in the GMF layer by performing element-wise operation on these latent features. The result from the GMF layer is trained iteratively by the deep neural network in the hidden layer to model the high-order interaction between the user and the item. Finally, the user's prediction ratings for the items are generated in the output layer, and a Top_N recommendation list based on the predicted rating is recommended to the user.

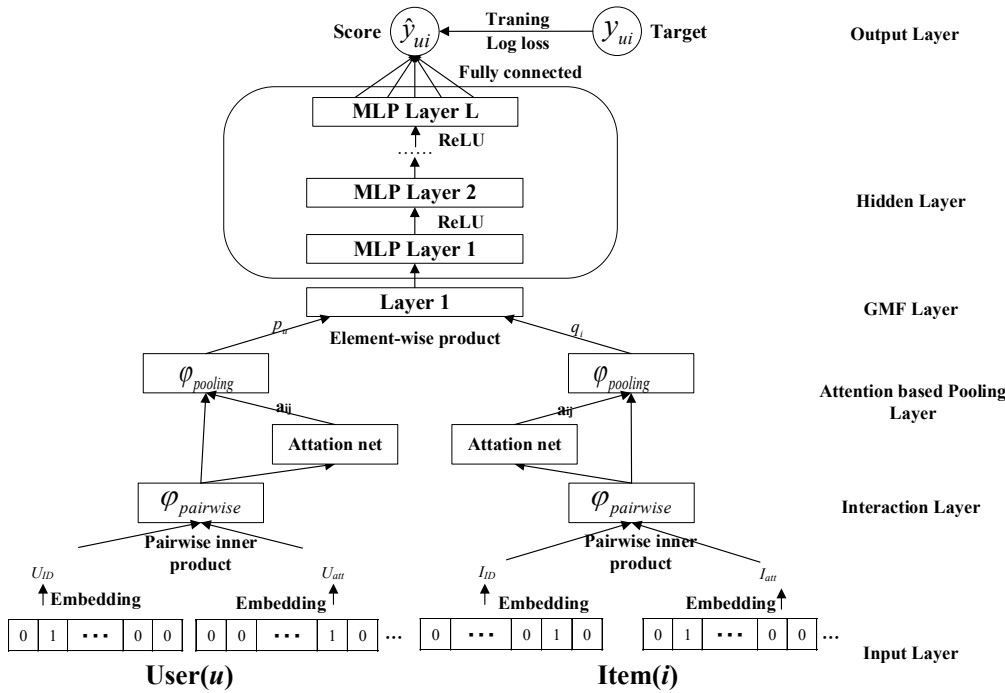

**Figure 1.** Information fusion-based deep neural attentive matrix factorization recommendation model structure.

### 3.2. Feature Crosses-Based Information Fusion

Traditional recommendation models often take rating information composed of explicit feedback data and interaction information composed of implicit feedback data as input, which leads to the model can only model partial user preferences and partial item representations because of the high sparsity of the data. Therefore, we introduce auxiliary information, namely the user's social-demographic features (age, gender, occupation, etc.) and the item's content features (categories, profiles, etc.) into the IFDNAMF model. Specifically, we utilize one-hot encoding to obtain the representations of category features (age, gender, ID, etc.). To solve the problem that the number of categorical features is large, which results in very sparse binary vectors, we adopt the embedding matrix to map the sparse and high-dimensional vector representations to the dense and low-dimensional space. Meanwhile, we adopt Fast-Text [34] to obtain the text attribute representations by averaging word embedding vectors. Similarly, we multiply the text attribute representations by the embedding matrix to map the sparse vector representations to the dense space. As introduced in Section 2, when conducting the auxiliary information fusion, most of the existing researches utilize the fully connected network to simply concatenate the features of different information, which can be regarded as the "or" operation in logic. The fully connected network cannot obtain cross-features and express the interaction between features, which makes the model ignore the "and" relationship between features. Considering a scenario in which ad click predictions are made for different users on the sites to recommend ads that meet the personalized needs of users. For the Disney advertisement on the sites, we assume that these features, namely the location: Shanghai, younger users, and the date: Friday, play a critical role in whether the user clicks it. Users who meet these features

are more likely to click on this ad when they see this ad. Intuitively, users matching the above features have a higher probability of clicking than users who only match one of them. In other words, the value of the information contained in the "and" relationship between the features is richer than the value of the information contained in the "or" relationship between the features. Ignoring this "and" relationship between features may result in the loss of much useful information in the modeling process, which may affect the performance of recommendation systems of the model.

Therefore, to introduce the information contained in the "and" relationship between features, we propose an information fusion method based on feature crosses in the IFD-NAMF recommendation model. The specific implementation is shown in the interaction layer in Figure 1. When fusing the auxiliary information, we adopt the element-wise product between different features to complete the interaction between features of different information domains in a targeted way and obtain different cross-features. That is, we perform the element-wise product between the user ID and each user attribute feature and the pairwise element-wise product between the user attribute features for the user, and we adopt the element-wise product to model the interaction between the item ID and each item attribute feature and the interaction between the item attribute features for the item. The pairwise element-wise product in Figure 1 is the operation of the element-wise product between the embedding vectors described above. Through the pairwise element-wise product between embedding vectors, we could obtain multiple cross-features of users and items, which is defined as:

$$\varphi_{pairwise}(u) = \{U_t \odot U_{t'} | t, t' \in \{ID, att_1, \ldots att_n\}, t \neq t'\} \tag{1}$$

$$\varphi_{pairwise}(i) = \{I_t \odot I_{t'} | t, t' \in \{ID', att'_1, \ldots att'_m\}, t \neq t'\} \tag{2}$$

where $U_t$ and $U_{t'}$ are the embedding vectors, including the user $ID$'s embedding vectors and the embedding vectors of user attribute features. $ID, att_1, \ldots att_n$ represents the user's $ID$ and $n$ attribute features. $I_t$ and $I_{t'}$ are the embedding vectors, including the item $ID$'s embedding vectors and the embedding vectors of item attribute features. $ID', att'_1, \ldots att'_m$ represents the item's $ID$ and $m$ attribute features.

Simultaneously, we compare the performance of the IFDNAMF model with the traditional recommendation model employing the fully connected network and demonstrate the effectiveness of our proposed IFDNAMF, which models the interaction between features of different information domains when conducting information fusion in Section 4.2.

### 3.3. Cross-Features Fusion Based on Attention Mechanism

After obtaining the cross-features of the user and the item through the interaction layer, we need to fuse these cross-features obtained by feature crossing of different information to finally obtain the user preference representation and the item representation, namely the potential feature vector of the user and the item, $p_u$ and $q_i$. However, it should be noticed that different cross-features have different degrees of importance on the final prediction result. Some cross-features may directly affect the final recommendation result. On the contrary, some cross-features are not important for the final result. If we combine these cross-features with the same weight without distinguishing the importance of these cross-features, these insignificant cross-features may become noise and offset the positive impact of the important cross-features on the final recommendation result, which ultimately affects the performance of recommendation model. Here, we still take the Disney advertising click prediction scenario as an example. The cross-feature obtained by the cross of age and date has a great influence on the click prediction result, and the cross-feature obtained by the cross of height and accent is irrelevant for users' clicks on Disney advertising. For two users, assuming that the cross-features representation obtained by the cross of their age and date are very different, the final click prediction result should be greatly different. However, if the two cross-features are fused with a weight of 1, the possible result is that the cross-feature obtained by the cross of height and accent makes up for the great difference

of the cross-feature obtained by the cross of age and date, which cause the two users' click prediction results are same. Therefore, it is necessary to distinguish the contribution of these cross-features on the final result by different importance levels. Here, we adopt the attention mechanism to learn the importance of different cross-features on prediction results, by which we fuse different cross-features. Specifically, by applying the attention net of the pooling layer based on the attention mechanism on the different cross-features, the attention scores of different cross-features of users and items $a_{ij}$ and $a'_{ij}$ are, respectively, learned, which represents the contribution of different cross-features to the prediction. The calculation formula is formulated as follows:

$$a_{ij}(U_t, U_{t'}) = \frac{e^{h^T ReLU(W(U_t \odot U_{t'})+b)}}{\sum_{U_t \odot U_{t'} \in \varphi_{pairwise}(u)} e^{h^T ReLU(W(U_t \odot U_{t'})+b)}} \tag{3}$$

$$a'_{ij}(I_t, I_{t'}) = \frac{e^{h^T ReLU(W(I_t \odot I_{t'})+b)}}{\sum_{I_t \odot I_{t'} \in \varphi_{pairwise}(i)} e^{h^T ReLU(W(I_t \odot I_{t'})+b)}} \tag{4}$$

where $W$ and $b$ represent the weight matrix and bias vector of the interaction layer to the attention net, respectively. $h$ represents the connection weight of attention net to the pooling layer.

Then, multiple cross-features of the user and the item are combined according to different contribution levels through the sum pooling operation, thereby obtaining the potential feature vector containing the attribute features of the user and the item, namely $p_u$ and $q_i$. The calculation process is formulated as follows:

$$p_u = \varphi(\varphi_{pairwise}(u, a_{ij}(U_t, U_{t'}))) = \sum_{t \neq t'} a_{ij}(U_t, U_{t'})U_t \odot U_{t'} \tag{5}$$

$$q_i = \varphi(\varphi_{pairwise}(i, a'_{ij}(I_t, I_{t'}))) = \sum_{t \neq t'} a'_{ij}(I_t, I_{t'})I_t \odot I_{t'} \tag{6}$$

### 3.4. GMF Structure Based on Multiple Hidden Layers

The traditional matrix factorization-based recommendation model cannot capture the complex non-linear interaction between users and items because of adopting the linear inner products. In response to this problem, many related works have been done, among which the GMF model is particularly conspicuous. By introducing a single layer neural network on the second-order interaction between the user and the item, the activation function and the bias term are employed to endow the model with the ability to model non-linear interaction between users and items. However, GMF is a shallow model, which can well model the low-order interaction features of user and item, it is not effective to capture the more abstractive features that contain more rich information, namely high-order interaction information. For example, if we enter the interaction information between the user and the music into the recommendation model with a single layer neural network, we could capture that the user prefers classical music. Then if we enter the obtained results into the next layer neural network for learning, we could get the result that the user tends to listen to Mozart's music in classical music. Therefore, we introduce the GMF structure based on multiple hidden layers in the IFDNAMF model, so that the model can not only model non-linear second-order interaction between users and items but also capture the high-order interaction between users and items. We can define the formulation as follows:

$$s_1 = \varphi(p_u, q_i) = p_u \odot q_i \tag{7}$$

$$Z_1 = \phi_1(s_1) = a_1(W_1^T s_1 + b_1) \tag{8}$$

$$Z_2 = \phi_2(Z_1) = a_2(W_2^T Z_1 + b_2)$$

$$......$$ (9)

$$Z_{L-1} = \phi_{L-1}(Z_{L-2}) = a_{L-1}(W_{L-1}^T Z_{L-2} + b_{L-1})$$

where $a_1$ is the activation function. $W_1$ and $b_1$ is the connection weight matrix and bias vector. $Z_{L-1}$ is the output of the $L-1th$ layer network. $a_2$ and $a_{L-1}$ are the activation functions of each hidden layer. $W_2^T$, $W_{L-1}^T$ and $b_2$, $b_{L-1}$ are the connection weight matrix and bias vector of each hidden layer, respectively.

## 4. Experiments

In this section, we conduct the experiments to demonstrate the feasibility and effectiveness of our proposed IFDNAMF recommendation model through the comparative analysis of the performance indicators with the baselines. The experimental analysis mainly focuses on the following three aspects:

(1)　The performance of recommendation systems of the IFDNAMF model;
(2)　The impact of the latent vector dimension on the performance of the model;
(3)　The impact of the number of hidden layers on the performance of the model.

### 4.1. Experiment Settings

#### 4.1.1. Datasets

We perform experiments with two public datasets: MovieLens (https://grouplens.org/datasets/movielens/1m/ (accessed on 26 September 2021)) and Book-crossing (http://www2.informatik.uni-freiburg.de/~cziegler/BX/ (accessed on 26 September 2021)) to verify the performance of the models.

The MovieLens is a movie dataset widely used in recommendation systems. It contains multiple versions with different data sizes. Here we use the MovieLens 1M dataset, referred to as MovieLens. The MovieLens contains 1 million rating records from 6000 users on 4000 movies, with scores ranging from 1 to 5. The user's attribute features include gender, age, job, and zip code. The movie's attribute features include the movie title and movie category.

Book-crossing is an open book dataset, which is composed of 278,858 users, 271,397 books, and 1,149,780 rating records, with a rating range from 1 to 10. The user's attributes include age and region. The book's attributes include title, author, publication year, and publisher. Since we adopt the leave-one-out evaluation strategy, in order to ensure that model evaluation can be carried out, each user needs to have at least one training data and one test data. Therefore, we filter out users and books with less than 5 interaction records. The filtered dataset contains more than 600,000 rating records of 21,915 users on 39,702 books. Although we filter the data, the data sparsity is still as high as 99.93%, so this is still a scenario with highly sparse datasets.

For each user, we sort all the interactions by timestamp, and select the last interaction record, which is the most recent interaction record and 99 non-interactive samples obtained by random sampling as the test sample to form the test set. The rest of the interaction records are regarded as the training sample to form the training set.

#### 4.1.2. Evaluation Protocol and Baselines

Because it is too time-consuming for each user to predict their preference on all non-interactive items, we follow a general strategy [12,18]. Through applying the sampling method described in Section 4.1.1, the test set composed of 1 positive sample and 99 negative samples is collected for each user, then the user's preference on these 100 items is predicted, after that the prediction rating is ranged from large to small to generate a top 10 ranking list. The performance of recommendation systems is measured by hit ratio (HR) and normalized discounted cumulative gain (NDCG) [35]. HR intuitively evaluates whether the test item is on the top 10 list. NDCG is used to measure the location of the positive test sample in the ranking list.



We compare IFDNAMF with the following models:

- **GMF:** GMF is an improved matrix factorization algorithm [7]. The matrix factorization model is generalized by the activation function and the connection weights with incomplete weights of 1 so that the model can model the non-linear second-order interaction between the user and the item;
- **GMF+MLP:** By removing the interaction layer, attention layer, and pooling layer of the IFDNAMF model, the IFDNAMF model can be generalized to the improved GMF model with the multi-layer hidden layer network, referred to as GMF+MLP, which is utilized to demonstrate the effectiveness of deep neural networks to model high-order interactions between the user and the item;
- **Concat:** The concat is the variant of the IFDNAMF model, which is the common method for traditional recommendation models to process the attribute information. This method completes the combination of features of information through the fully connected layer, which ignores the interaction between different feature domains, and lacks pertinence;
- **Sum pooling:** To verify the effectiveness of the method that endows the cross-features with different weights by the attention mechanism, the IFDNAMF model is compared with the model that contains the same network structure with the IFDNAMF model removing the attention layer, referred to as sum pooling, which crosses the features by element-wise product and combines the cross-features by sum pooling operation.

We implemented the above models and IFDNAMF recommendation model based on Tensorflow. For the neural network model, the model is initialized by a truncated normal distribution with a mean of 0 and a variance of 0.01. The model is optimized by the cross-entropy loss function and the Adam gradient descent algorithm. The batch size and initial learning rate are 256 and 0.001, the activation function of the hidden layer and the output layer are ReLU and Sigmoid, respectively.

### 4.2. Experimental Results and Analysis

We conduct the experiments to demonstrate the feasibility and effectiveness of our proposed model through the comparative analysis of the performance indicators with the baselines from the following three aspects.

#### 4.2.1. Result 1: Performance Comparison

To explore the performance of IFDNAMF, we compared the performance of the proposed IFDNAMF model with the four models: GMF, GMF + MLP, concat, sum pooling, and GMF. The five models were trained and tested on MovieLens and Book-crossing, respectively. Table 1 shows the performance of these models's HR@10 and NDCG@10.

**Table 1.** The performance of the five models on MovieLens and Book-crossing

| Dataset | MovieLens | | Book-Crossing | |
|---|---|---|---|---|
| **Baseline** | **HR@10** | **NDCG@10** | **HR@10** | **NDCG@10** |
| **GMF** | 0.7141 | 0.4357 | 0.6518 | 0.3739 |
| **GMF+MLP** | 0.7318 | 0.4502 | 0.6831 | 0.4150 |
| **concat** | 0.7369 | 0.4528 | 0.6878 | 0.4157 |
| **Sum pooling** | 0.7450 | 0.4600 | 0.6953 | 0.4212 |
| **IFDNAMF** | 0.7501 | 0.4633 | 0.6984 | 0.4229 |

It is not hard to see from Table 1 that on both datasets, the performance of the two indicators HR@10 and NDCG@10 is: IFDNAMF > sum pooling > concat > GMF+MLP > GMF. The performance of our proposed IFDNAMF model is superior to the baselines on both datasets. Compared with the sum pooling model, the two indicators of the IFDNAMF model on the Book-crossing have increased by 0.31% and 0.17%, while the two indicators of the IFNDAMF model on MovieLens have improved by 0.51% and 0.33%. Compared with

the GMF model, the HR@10 and NDCG@10 of IFDNAMF have increased by 3.6% and 2.67% on MovieLens, while 4.66% and 4.9% on the Book-crossing. Among these models, GMF performs the worst, followed by the GMF+MLP model. This may be caused by the fact that the GMF and GMF+MLP only model the user ID feature and item ID feature, which leads to the loss of the partial user preference and item representation, that is, they could only represent the partly information of user preference and item representation. The other three models have fused features of other attribute information of users and items, as well as the user ID feature and the item ID feature. Through the auxiliary information, they could model user preference and item representation more comprehensively and accurately from multiple aspects, thereby the performance of recommendation systems gets improved. This also demonstrates the effectiveness of the way that introduces auxiliary information into the model to more comprehensively model user preference and item representation.

Furthermore, compared with the sum pooling model and IFDNAMF model, the performance improvement of the concat model is relatively worse. This indicates that, when fusing attribute information, the simple concatenating operation on the segmented attribute feature vectors through the fully connected network is not effective to capture the interactive relationships between the features, which will lead to the limitation on the performance of the model, while through the element-wise product, the cross between different feature domains is completed in a targeted manner, and the cross information between features will be captured, which enhances the model's ability to represent different data patterns. Meanwhile, Table 1 shows that the IFDNAMF model performs better than the sum pooling model on both datasets. This is due to that different cross-features have different degrees of importance in the rating prediction task, compared with the sum pooling model, the IFDNAMF model can assign different weights to different cross-features through the attention mechanism, thereby effectively distinguishing the correlation of different features on the result. This indicates that it is feasible and effective to introduce the attention mechanism to the model. In addition, the HR@10 and NDCG@10 of the GMF+MLP perform better than the GMF model on both datasets. This is caused by the fact that the GMF is a shallow model, although it can well model the non-linear second-order interaction between users and items, it cannot effectively model the high-order interaction between users and items. This indicates that by adding hidden layers, the model could capture more abstractive information between users and items to model high-order interactions through the deep network.

Furthermore, because we split the training and test sets by random sampling, we perform the statistical significance tests. We repeat the experiment 20 times with random sampling, and the final experimental result was obtained by averaging the results of 20 experiments. Here, we use the no-repeat two-way ANOVA method to test the difference significance of the experimental results of GMF and IFDNAMF model under different datasets with different metrics, respectively. The results are showed in the Tables 2 and 3. The difference significance test results are explicit that there are significant differences in the experimental results on different datasets and different metrics, and the differences are extremely significant.

**Table 2.** The results of statistical significance tests with HR and NDCG on MovieLens

| Metrics | HR | | | NDCG | | |
|---|---|---|---|---|---|---|
| **Difference** | **F** | ***p*-Value** | **F Crit** | **F** | ***p*-Value** | **F Crit** |
| **row** | 6013.799 | $3.1 \times 10^{-25}$ | 4.38075 | 0.809181 | 0.67545 | 2.168252 |
| **column** | 43481.85 | $2.18 \times 10^{-33}$ | 4.38075 | 0.557571 | 0.893962 | 2.168252 |

**Table 3.** The results of statistical significance tests with HR and NDCG on Book-crossing

| Metrics | HR | | | NDCG | | |
|---|---|---|---|---|---|---|
| **Difference** | **F** | ***p*-Value** | **F Crit** | **F** | ***p*-Value** | **F Crit** |
| **row** | 7990.425 | $2.1 \times 10^{-26}$ | 4.38075 | 1.080408 | 0.433948 | 2.168252 |
| **column** | 649512 | $1.53 \times 10^{-44}$ | 4.38075 | 1.986402 | 0.071824 | 2.168252 |

4.2.2. Result 2: The Impact of the Latent Vector Dimension on the Performance

To explore the impact of the latent vector dimension on the performance of the model, we compared the performance of the four models: GMF+MLP, concate, sum pooling, and IFDNAMF under the different latent vector dimensions of 10, 20, 40, and 80 on MovieLens and Book-crossing. The results are shown in Figures 2 and 3.

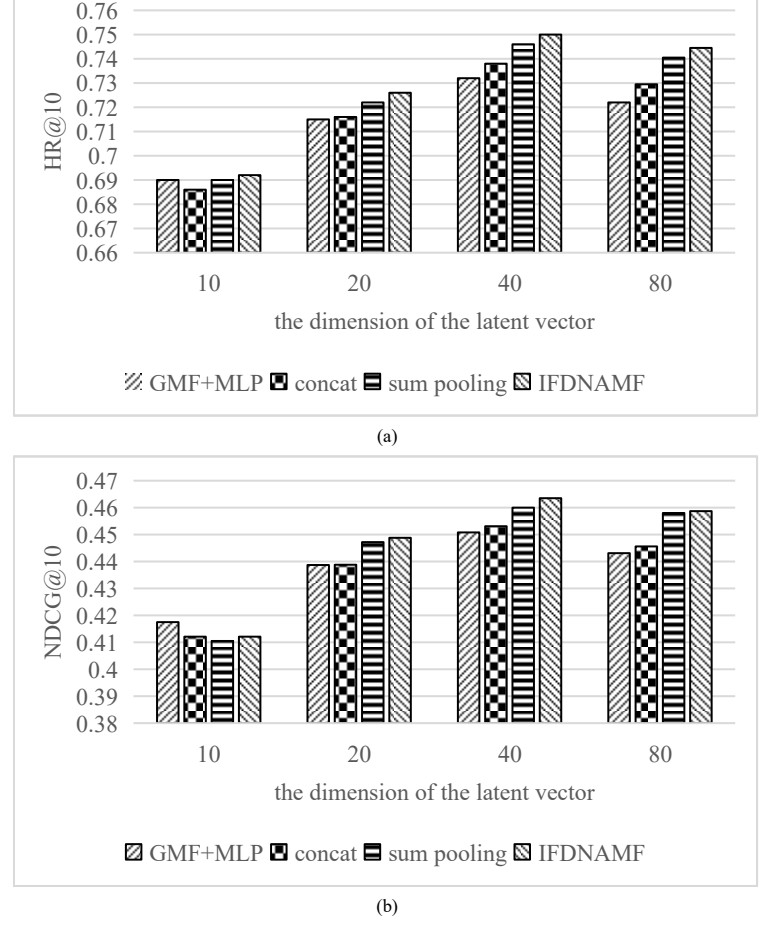

**Figure 2.** The HR@10 and NDCG@10 of models with different potential vector dimension on MovieLens. (**a**) HR@10. (**b**) NDCG@10.

Figures 2 and 3 show that with the continuing increase in the potential dimensions, the HR@10 and NDCG@10 of the four models increase continuously, and the performance of recommendation systems of the models gets improve continuously. This is because before reaching the optimal latent vector dimension, as the dimension increases, the latent vector could contain more information, which assists the model in describing the user features and item features more comprehensively and specifically to model the latent features of users and items better, so the performance of recommendation systems is better. However, the dimension should not be too large. Because when the latent vector dimension is too large, it may lead to over-fitting. This is why when the latent vector dimension reaches

40 on the MovieLens, the HR@10 and NDCG@10 of models begin to decline, and when the latent vector dimension reaches 20 on the Book-crossing, HR@10 and NDCG@10 also begin to decline. Meanwhile, we can conclude from Figures 2 and 3 that the optimal latent vector dimension of the IFDNAMF model is 40 on MovieLens and 20 on Book-crossing. Furthermore, in different dimensions (20, 40, and 80), compared to the GMF+MLP model, the performance of the sum pooling model and IFDNAMF model which cross different features by element-wise product achieve great improvement, while the performance of the concat model which concatenates the features by the fully-connected network gets limited improvement. This demonstrates that, when fusing attribute information, it is an effective way to simulate the interaction of features by element-wise product, while combining the segmented feature vectors through the fully-connected network is not effective to represent the features, so the improvement of model performance is small. In addition, in different dimensions (20, 40, and 80), compared with the sum pooling model, the HR@10 and NDCG@10 of the IFDNAMF model introducing the attention mechanism increase, and the model performance has been further improved. This indicates that not all cross-features have the same degree of association with the result, and by assigning the cross-features with different weights, the model can obtain the ability to adjust the feature weights according to the characteristics of the sample, which is significative to improve the prediction ability of the model.

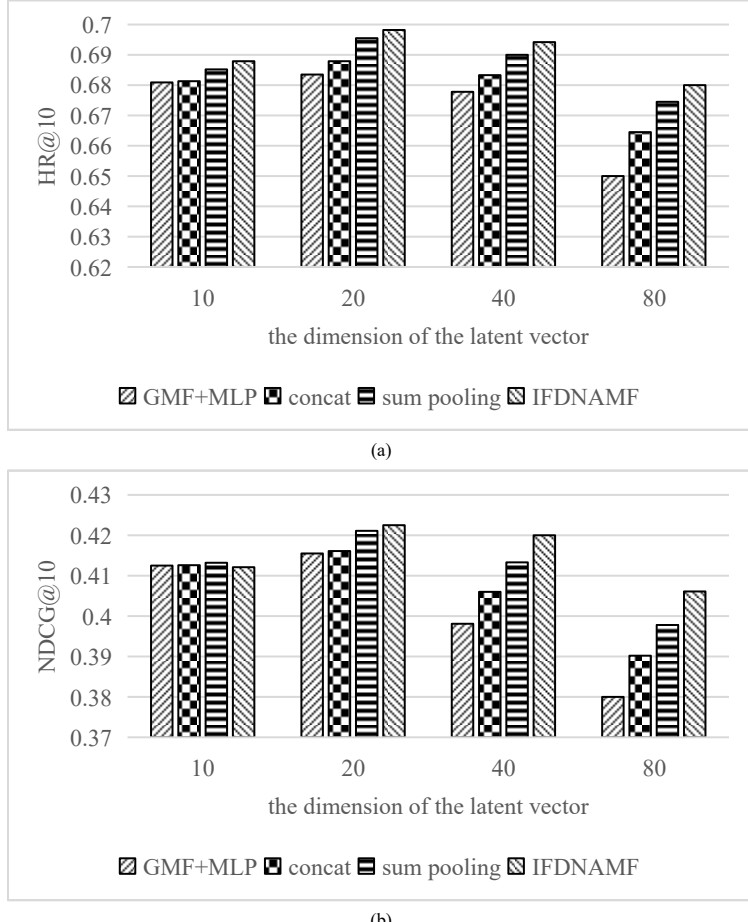

(a)

(b)

**Figure 3.** The HR@10 and NDCG@10 of models with different potential vector dimension on Book-crossing. (**a**) HR@10. (**b**) NDCG@10.

### 4.2.3. Result 3: The Impact of the Number of Hidden Layers on the Performance

In Section 4.1.2, we generalize the IFDNAMF model to the GMF+MLP model by removing the interaction layer, attention layer, and pooling layer. Through the experiments and analysis in Result 1, it is not difficult to conclude that by adding multi-layer neural network to the GMF model, the model is endowed with the ability that models the high-order interaction between the user and the item, so the model could learn more abstractive features, and then the model obtains a better ability of expression. To explore the influence of the number of hidden layers on the performance of recommendation systems of the model, we set up control group experiments with hidden layers of 1, 2, 3, and 4, respectively. The model with one layer of neural network is denoted as MLP-1, MLP-2 represents the model with two layers of neural network, etc. We conducted the experiments on MovieLens and Book-crossing and got the HR@10 and NDCG@10 of the four models: GMF+MLP, concat, sum pooling, and IFDNAMF of the four control groups experiments under the optimal potential vector dimension, respectively. The results are shown in Figures 4 and 5.

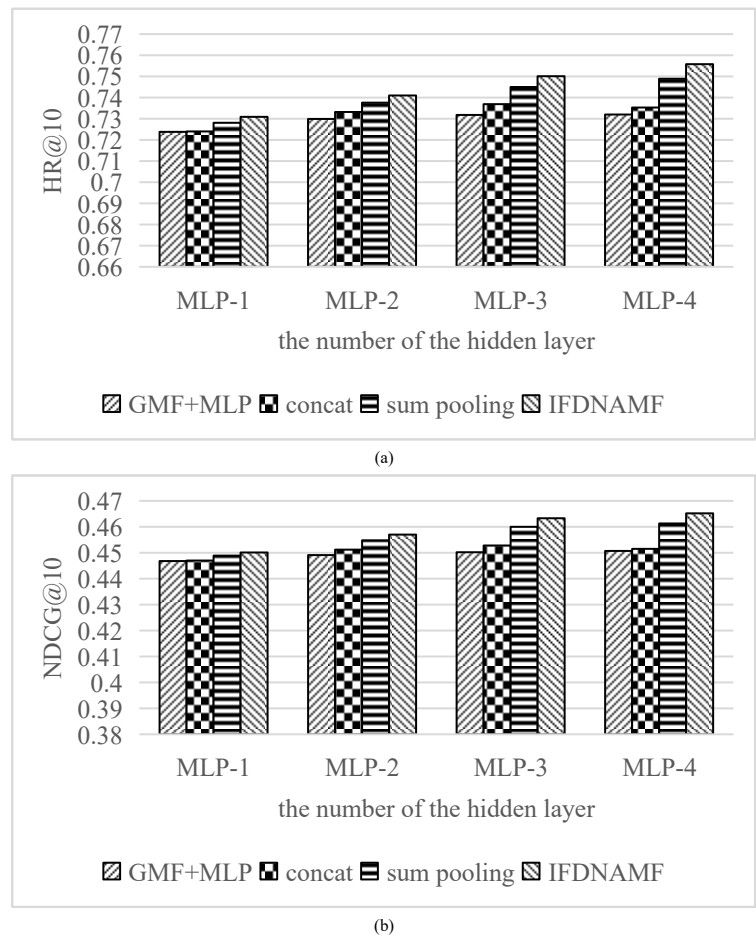

**Figure 4.** The HR@10 and NDCG@10 of models with different layers on MovieLens. (**a**) HR@10. (**b**) NDCG@10.

Figures 4 and 5 show that as the number of hidden layers increases continuously from 1, the HR@10 and NDCG@10 of the model are constantly growing, and the model's performance gets significantly improved. This is because the single layer neural network can map input data to another abstract space to learn more abstractive features, and by increasing the number of hidden layers, the model could learn richer and more abstractive information to model the interaction between users and items better. In addition, through introducing hidden layers, the model obtains the ability to learn non-linearity, which

improves the model's ability of expression. This indicates that by adopting the deep neural network, the IFDNAMF model could learn high-order interaction between users and items, thereby the model's performance of recommendation systems gets improved further. However, as the number of layers of the deep neural network increases further, the HR@10 and NDCG@10 increase slowly, even when the number increases from 3 to 4, the HR@10 and NDCG@10 of the concat model on MovieLens begin to decline, which indicates that it is not that the more layers the MLP has and the deeper the neural network is, the better the model performance of recommendation systems is. This is because too many hidden layers will lead to over-fitting, which will limit the performance of recommendation systems of the model. Meanwhile, due to too many hidden layers in the neural network, the model parameters will increase exponentially, which will increase the difficulty to train and make it difficult for model to converge.

In addition, the HR@10 and NDCG@10 of the models with attribute features, namely the concat model, sum pooling model, and IFDNAMF model, are higher than the GMF+MLP model, which further indicates that the introduction of auxiliary information is beneficial to the performance of the model. Furthermore, it can be seen that the sum pooling model performs better than the concat model, which connects feature vectors through concatenating operation. Further, the IFDNAMF model performs best. This also verifies that it is meaningful to improve the performance of the model by explicitly completing the crossover between feature domains through the element-wise product and assigning different weights to the cross-features through the attention mechanism.

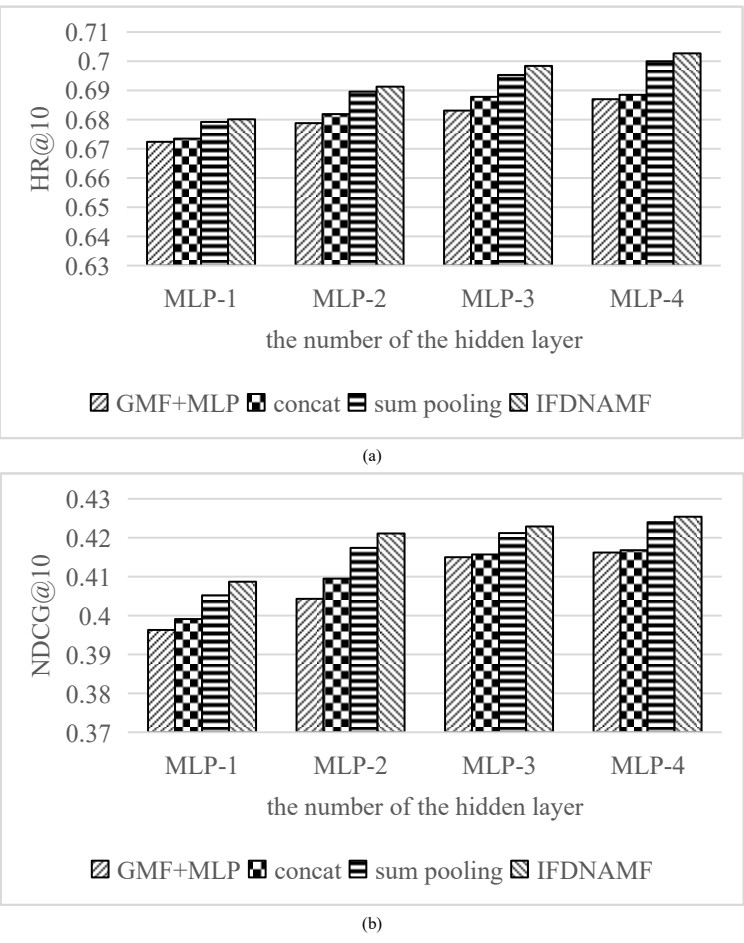

**Figure 5.** The HR@10 and NDCG@10 of models with different layers on Book-crossing. (**a**) HR@10. (**b**) NDCG@10.

## 5. Conclusions

In this paper, we introduce the attribute features of users and items to the matrix factorization recommendation system, and adopt the element-wise product operation between features of different information domains to model the cross-features, which could overcome the limitation of the fully connected network and model the user preference and item representation more comprehensively and accurately. In addition, we also utilize the attention mechanism to distinguish the importance of different cross-features on the prediction results. Meanwhile, we utilize the structure that adds the hidden layer on GMF to solve the problem of the linear inner product of matrix factorization and model the non-linear high-order interaction between the user and the item. Meanwhile, we conduct extensive experiments on two datasets: MovieLens and Book-crossing, the results demonstrate the feasibility and effectiveness of the model. Experimental results demonstrate the feasibility and effectiveness of the model.

The element-wise product is an element-by-element multiplication between vectors, which may ignore the interaction between elements of different dimensions of vectors. Therefore, in the future, we may attempt to adopt the outer product to simulate the interaction between features. Meanwhile, the IFDNAMF does not take into account the neighbor information. Thus, introducing neighbor information into the IFDNAMF may be explored in the future.

**Author Contributions:** Conceptualization, Z.T. and L.P.; methodology, Z.T. and L.P.; validation, L.P.; formal analysis, Z.T., L.P., P.Y., and R.W.; investigation, Z.T.; resources, P.Y.; data curation, L.P.; writing—original draft preparation, Z.T.; writing—review and editing, Z.T., P.Y., and R.W.; visualization, L.P.; supervision, R.W.; project administration, R.W.; funding acquisition, R.W. All authors have read and agreed to the published version of the manuscript.

**Funding:** This research was funded by the Scientific Technological Innovation Foundation of Shunde Graduate School, USTB under Grant No. BK19CF010 and BK20BF012, and in part by the National Natural Science Foundation of China under Grant No. 62173158 and No. 72004147.

**Institutional Review Board Statement:** Not applicable.

**Informed Consent Statement:** Not applicable.

**Data Availability Statement:** Not applicable.

**Conflicts of Interest:** The authors declare no conflicts of interest.

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
