# Peer review of "Information Fusion-Based Deep Neural Attentive Matrix Factorization Recommendation"

_algorithms, doi:10.3390/a14100281_

Round 1
Reviewer 1 Report
The paper proposes a recommendation model, which introduces the attribute information to the model and adopts the element-wise product between different information domains to learn the cross-features. The deep neural network is used to learn the high-order interaction between users and items. The results of experiments on two benchmark datasets are presented. The article should be improved both in terms of methodology as well as presentation by considering the comments and suggestions provided below.
Major comments:
- What is the knowledge gap bridge by this study? The novelty of this study is not clear as there were numerous matrix factorization based recommendation schemes introduced before. The novelty and contribution of this work must be clearly stated at the end of the Introduction section.
- The background and motivation part of the paper should be extended by discussing more previous works while discussing their limitations and shortcomings as a motivation for this study. Moreover, some of the references are outdated (some are more than 10 years old), which does not reflect the progress in this rapidly evolving area of research. I would suggest to discuss more recent state-of-the-art publications such as “Enriching Non-negative Matrix Factorization with Contextual Embeddings for Recommender Systems”, “Efficient neural matrix factorization without sampling for recommendation”, “Intelligent recommendation of related items based on naive bayes and collaborative filtering combination model”, and “Recommendation Based on Review Texts and Social Communities: A Hybrid Model”, among others.
- Line 240: you used one-hot encoding to obtain the representations of category features. However, when the number of categorical features is large, it may lead to feature space explosion. How do you deal with this problem? Did you consider any alternatives to one-hot encoding?
- Lines 248-249: “The fully connected network cannot obtain cross-features and express the interaction between features” – Can you support these considerations by a longer discussion and/or appropriate references?
- How do you avoid overfitting by your neural network model? What is the training stoppage criterion? More details about the training of the neural network should be provided for replicability.
- The Hit-ratio and Normalized Discounted Cumulative Gain (NDCG) metrics have been used for performance evaluation. I suggest to add more performance metrics such as F1 score of item recommendation. The evaluation should be done not only in terms of accuracy, but also in terms of recommendation diversity, as was done in “Customer Satisfaction of Recommender System: Examining Accuracy and Diversity in Several Types of Recommendation Approaches”.
- Compare your results with the results of other authors obtained on the same datasets.
- Discuss the limitations of the proposed model and threats-to-validity of the experimental results.
- Improve the conclusions. Use the main numerical results from your experiments to support the claims.
Minor comments:
- Equations 1-7: explain the meaning of dot operation.
- Increase the size of Figures 2 and 3 for better readability. I also suggest to increase text font size in figures for better readability.
- Figures 8 and 9 are too small and the text is almost unreadable.
Reviewer 2 Report
In this submission, the authors present a deep neural matrix factorization method for recommender systems, outlining the details of their approach and evaluating it on two popular recommendation datasets. In my opinion, the following issues need to be addressed prior to considering this work for publication.
1. l75-76 “connection weight of nto all 1”. It is not clear what the authors mean here. Please elaborate more on this phrase.
2. l99-100. It is not true that it is difficult to integrate other forms of data in ratings’ matrix factorization. In fact, many works have just followed this course of action, with interesting results. In this respect, the authors should cite the following works in their related work section.
Chen, C., Zeng, J., Zheng, X., & Chen, D. (2013, September). Recommender system based on social trust relationships. In 2013 IEEE 10th International Conference on e-Business Engineering (pp. 32-37). IEEE.
Wei, S., Zheng, X., Chen, D., & Chen, C. (2016). A hybrid approach for movie recommendation via tags and ratings. Electronic Commerce Research and Applications, 18, 83-94.
Alexandridis, G., Siolas, G., & Stafylopatis, A. (2017). Enhancing social collaborative filtering through the application of non-negative matrix factorization and exponential random graph models. Data Mining and Knowledge Discovery, 31(4), 1031-1059.
Alexandridis, G., Tagaris, T., Siolas, G., & Stafylopatis, A. (2019, May). From free-text user reviews to product recommendation using paragraph vectors and matrix factorization. In Companion Proceedings of The 2019 World Wide Web Conference (pp. 335-343).
3. l241: One-hot encoding is counter-intuitive, especially of user and item IDs. Given that in typical recommender systems those sets are in the order of magnitude of thousands (as, for example, in the datasets of the experimental section), this would result in very sparse binary vectors. I tend to think that the attention mechanism introduced in the next layer actually tries to ameliorate the negative side effects of the chosen encoding. Therefore the authors need to devise a more compact encoding scheme.
4. l242-243 “we perform convolution operation on the word vector matrix of the short text”. This sentence is vague; do you actually employ pre-trained neural language embeddings (e.g. word2Vec, GloVe) here?
5. l.328, Equations 7-9, describing how deep neural networks work are already well-known and therefore provide insignificant contribution to the manuscript. In my opinion, they should be removed.
6. l352 “114,9780” -> “1,149,780”
7. l353-356. Removing cold start users and items from the dataset is counter-intuitive and an indication that the proposed model cannot deal with the cold-start problem and data sparsity in general, as implied in the abstract of this submission.
8. l358-359. The splitting of the dataset in training and test sets is not clear and should be explained in more detail.
9. l407. The results presented on Table 1 and on Figures 2-5 are based on the random split of data in training and test sets discussed in Section 4.1. Therefore, they should be assessed by appropriate statistical significance tests.
10. l452 & l482. Enlarge the subfigures of Figures 2-5 by placing them one under the other (and not one next to the other)
Round 2
Reviewer 1 Report
The paper was improved, but still, several issues remain.
- The overview of related works still discusses some outdated works (some over 10 years old). Please update this part of the paper to reflect state-of-the-art.
- Table 1: the results of all five models are very close. Is the difference between model performance statistically significant? Which model has the best performance? Perform statistical testing using a non-parametric ranking test such as Friedman test, and post-hoc Nemenyi test. Represent the mean ranks using Critical distance diagrams. Discuss the results.
- Evaluate recommendation diversity and present the values of appropriate metrics.
- Figures 4, 5: the plots are confusing. Usually, line plots are not recommended when the dependent variable values are not related sequentially. Use for example bar plots instead.
- Compare your results with the results of other authors achieved on the same benchmark datasets.
- There are some grammar errors and typos. Language polishing and careful text proofing are required.
Reviewer 2 Report
After reading both the revised version of the manuscript and the author response file, a number of issues still need to be addressed.
- Response 1: include the explanatory text in the manuscript
- Response 2: include the 4 suggested references in the manuscript
- Response 3: explicitly mention the problem of the sparsity of one hot encoding and how it is addressed in your proposed architecture
- Response 4: make an explicit reference to the pre-trained text convolutional neural networks you've used, giving the appropriate citation as well
- Response 7: add this explanation on data filtering in the appropriate lines
- Response 9: include the statistical significance tests as an extra table to the manuscript
Round 3
Reviewer 1 Report
The quality of the paper has improved. The statistical analysis increased the value of the paper, while the figures have become better understandable. The paper can be accepted now.